# Social robots in research on social and cognitive development in infants and toddlers: A scoping review

**Solveig Flatebø**[1]*, **Vi Ngoc-Nha Tran**[2], **Catharina Elisabeth Arfwedson Wang**[1], **Lars Ailo Bongo**[2]

1 Department of Psychology, UiT The Arctic University of Norway, Tromsø, Norway, 2 Department of Computer Science, UiT The Arctic University of Norway, Tromsø, Norway

* solveig.flatebo@uit.no

**Data Availability Statement:** All data are available from the OSF database doi.org/10.17605/OSF.IO/WF48R.

**Funding:** The author(s) received no specific funding for this work.

## Abstract

There is currently no systematic review of the growing body of literature on using social robots in early developmental research. Designing appropriate methods for early childhood research is crucial for broadening our understanding of young children's social and cognitive development. This scoping review systematically examines the existing literature on using social robots to study social and cognitive development in infants and toddlers aged between 2 and 35 months. Moreover, it aims to identify the research focus, findings, and reported gaps and challenges when using robots in research. We included empirical studies published between 1990 and May 29, 2023. We searched for literature in PsychINFO, ERIC, Web of Science, and PsyArXiv. Twenty-nine studies met the inclusion criteria and were mapped using the scoping review method. Our findings reveal that most studies were quantitative, with experimental designs conducted in a laboratory setting where children were exposed to physically present or virtual robots in a one-to-one situation. We found that robots were used to investigate four main concepts: animacy concept, action understanding, imitation, and early conversational skills. Many studies focused on whether young children regard robots as agents or social partners. The studies demonstrated that young children could learn from and understand social robots in some situations but not always. For instance, children's understanding of social robots was often facilitated by robots that behaved interactively and contingently. This scoping review highlights the need to design social robots that can engage in interactive and contingent social behaviors for early developmental research.

## Introduction

Early childhood encompasses the infant and toddler years, marked by gradual but rapid growth in both social and cognitive development [1, 2]. Social development involves acquiring skills to interact and build social bonds with others, whereas cognitive development refers to developing skills related to thinking and reasoning processes [1, 2]. Research in these two

**Competing interests:** The authors have declared that no competing interests exist.

subdisciplines focuses on a diverse range of abilities, such as attachment [3], imitation [4], play [5, 6], memory [7], theory of mind [8], social cognition [4], and language acquisition [9, 10]. Theory of Mind (ToM), the ability to attribute underlying mental states like beliefs, desires, and intentions to others [11–13], has not previously been studied in pre-verbal infants [14, 15]. However, recent advances in methods have demonstrated that a rudimentary ToM may emerge earlier than the traditional assumption at the age of four [14, 15]. In line with this research, an interesting question is whether infants attribute mental states to non-human agents. Similarly, animacy understanding, the ability to classify entities as animate or inanimate [16–18], has been demonstrated in infants as young as two months [19–22], and by three years of age, children are good at understanding this distinction. Research on animacy examines how young children distinguish living beings and objects based on featural and dynamic cues such as faces, contingency behavior, and goal-directed or self-generated movement, which may involve using non-human agents possessing such cues [16, 23–27].

Developmental psychology uses diverse methodologies, designs, data-gathering instruments and materials, and formats for stimuli presentation, and the research can be conducted in various research settings [28]. Using social robots as part of research methods has emerged as a promising way to gain social and cognitive developmental insights [29–31]. Some pioneering studies have also demonstrated that social robots can contribute to cognitive assessments of elderly people and children with autism [32, 33]. These robots are designed for social interactions with humans, and they are often physically embodied, with human or animal-like qualities, and can be autonomous or pre-programmed to perform specific actions, and they engage in social interactions [34, 35]. Social robots often have an anthropomorphic design with human-like appearance and behavior. For example, they commonly have heads with facial features and can display various social behaviors such as facial expressions, eye contact, pointing, or postural cues [36–38]. Two social robots commonly used for research on social and cognitive development skills are Robovie [39] and NAO [40]. In research settings, social robots can serve various roles, such as social partners in interactions [e.g., 40, 41], teaching aids delivering learning content [40, 42, 43], and they can be equipped with sensors and cameras to record child behaviors [39].

There are several research advantages of using social robots that are not easily achievable through other means when studying young children. Firstly, they provide a level of control and consistency that can be challenging to achieve with human experimenters [32, 44]. Secondly, because social robots are designed for social interactions, they might have potential in research on social learning situations such as imitation studies. Third, the socialness of robots in appearance and behavior [45], in addition to their novelty, make them potentially more suited to capture a child's attention and sustain their engagement over longer time periods for a variety of testing purposes. Lastly, social robots offer a compelling avenue for advancing our understanding of young children's early ToM and animacy understanding related to non-human agents with rich social properties and how they represent social robots specifically.

## The current review

Although social robots are increasingly used in various settings with children, little is known about their utility as a research tool investigating social and cognitive concepts in infants and toddlers. We need to determine at which stages in early childhood children are receptive to and can learn from these robots. Currently, there is no available scoping review or systematic review of the available body of literature in this field. A review of the existing literature is needed to advance our understanding of social robots' relevance in research with younger age groups and map the current state of knowledge in this field. Given the potential diversity in

methodologies, research designs, and the wide range of developmental topics and concepts in the present research field, we decided to do a scoping review. Consequently, the main objective of the current scoping review is to provide a comprehensive overview and summary of the available literature on the use of social robots as research tools for studying the social and cognitive development of typically developing infants and toddlers aged 2 to 35 months.

Our focus is on research using social robots to inform child development, rather than research exclusively focusing on robot skills and application. We focus on typically developing children in the infancy and toddler years, younger than 3 years. We exclude neonates (0–2 months) and preschoolers (3–5 years) due to the notable distinctions in their developmental stages, which may necessitate different research methods compared to those used for infants and toddlers. Our definition of social robots is broad, encompassing all embodied robots exposed to children in a research context, irrespective of form and presentation format. However, we recognize the significance of eyes in early childhood communication [46] and, consequently, restrict our inclusion to only robots featuring eyes. Our definition covers both robots commonly defined as social robots as well as robots with social features in form and/or behavior. We chose this definition because both types of robots might be relevant for how non-human agents with richer social features can inform social and cognitive development.

This review will provide an overview of the research literature, covering research on concepts of social and cognitive development using robots, the research methods employed, and the types of robots used and their purposes. Also, our aim is to summarize the research trends by identifying the primary research focuses and findings. Finally, we want to summarize the reported gaps and challenges in this research field. Hopefully, the current review can be valuable for future research, helping to decide how to employ social robots in research settings with infants and toddlers and to support the development of age-appropriate robots for children.

## Method

We conducted a scoping review, which aimed to explore and map the concepts and available literature in a given field of research [47]. Like systematic reviews, scoping reviews follow rigorous and transparent methods [47, 48]. But, differently from systematic reviews, scoping reviews ask broader rather than specific research questions to encompass the extent and breadth of the available literature of a given field [47, 48]. We used The Preferred Reporting Items for Systematic Reviews and Meta-Analyses Extension for Scoping Reviews (PRISMA-ScR) (S1 Checklist) to improve this scoping review's methodological and reporting quality. We preregistered the protocol for this study on Open Science Framework on May 19, 2023 (see updated version of the protocol: https://osf.io/2vwpn/). We followed the recommendations of the Johanna Briggs Institute (JBI) [49] and the first five stages in the methodological framework of Arksey and O'Malley [47] and Levac and O'Brien's advancements of this framework [50].

### Stage 1: Identifying the research questions

The review was guided by three research questions: 1) What is the extent and nature of using social robots as a research tool to study social and cognitive development in infants and toddlers? 2) What are the primary research focus and findings? 3) What are the reported research gaps and challenges when using social robots as a research tool?

### Stage 2: Identifying relevant studies

**Inclusion criteria.** We developed inclusion criteria related to the publication type, target child population, the robot type, and the research focus (Table 1) to focus the scope of the review.

**Table 1. Inclusion criteria.** In the full-text screening, we excluded studies by the first unmet inclusion criteria, i.e., we checked if the publication met the criteria for publication type first, then for the target population, robot type, and finally, the research focus.

| Criterion | Included |
|---|---|
| **1. Publication type** | |
| Time frame | 1990 until May 29/05/2023 |
| Availability | Full texts available through open access or through our university subscription |
| Publication type | Peer-reviewed journal articles, journal magazine articles, preprints, and conference proceedings with full papers for empirical studies |
| Language | English |
| Research methodology | Empirical studies using quantitative, qualitative, or mixed methods |
| **2. Target child population** | |
| Participants | Publications with an exclusive focus on typically developing children between 2 to 35 months of age |
| **3. Robot type** | |
| Robot | Humanoid or non-humanoid form. Embodied robots, including partly animated robots. Full or partly autonomous. The robot must have eyes. The robot can be physically or virtually present in the child's environment, either as a physical robot or appearing in a video. The authors of the studies do not need to define the robot as social |
| **4. Research focus** | |
| Focus | Focus on child development, i.e., the robot is used to assess social and/or cognitive development in children. The publication includes an experiment, a pilot study, or a trial to test social and/or cognitive child development |

We consulted multiple databases to identify studies, as social robotics is an interdisciplinary field. We included conference proceedings and preprints because studies within robotics are often published in this format [51–53].

## Search strategy

We searched for literature in PsychINFO (OVID), Education Resources Information Center (ERIC, EMBASE), and Web of Science. We searched for preprints using the Preprint Citation Index in Web of Science and in PsyArXiv. All searches were done on 29 May 2023. In consultation with an academic librarian, we developed a search strategy and search terms, which are presented in the S1 File. We used controlled vocabulary in addition to keywords when searching in PsychINFO and ERIC. Web of Science and PsyArXiv lack their own controlled vocabulary, so PsychINFO and ERIC keywords were used in the searches. We categorized the search terms into three categories: robot type, target child population, and social and cognitive developmental concepts. For a comprehensive search, we used the search terms "robot*", "robotics", "social robotics", and "human robot interaction" related to robot type category. Moreover, for the target child population category we used terms like "infan*", "toddler*", "child*", "infant development", and "childhood development". Lastly, for developmental concepts we used terms such as "cognitive development", "social development", "social cognition", and "psychological development".

## Stage 3: Study selection

We developed a screening questionnaire a priori (doi.org/10.17605/OSF.IO/4BGX6), which all reviewers (SF, LAB, and VT) piloted initially on a random sample of studies. After revising the screening questionnaire, we started screening studies for eligibility in the web-based software Covidence [54]. We removed duplications manually and by using the Covidence duplicate

check tool. All studies were screened by two reviewers independently using the screening questionnaire. The first author (SF) screened all studies, whereas LAB and VT screened half of the studies each. We resolved disagreements by team discussion. The studies were screened through a two-step process: 1) screening of titles and abstracts; 2) screening of full texts. In full-text screening, we followed the exclusion reason order in Table 1 and excluded studies by the first unmet inclusion criteria.

### Stage 4: Data charting

We developed a data charting template a priori in Covidence and we used it to chart data from the studies included. The first author (SF) piloted the data charting template on five studies and iteratively modified it based on recommendations [50]. The main revisions included changes to the template layout, adding entities (i.e., final sample size and physical CRI contact), and providing more charting instructions and explanations of the entities. The details about the newest version of the charting template and charted entities are available at OSF (doi.org/10.17605/OSF.IO/B32R6). The first author (SF) charted data from each publication, and a second reviewer (LAB or VT) checked the charted data for completeness and accuracy in Covidence. Disagreements were resolved by discussion in the research team. We charted data regarding general study characteristics (e.g., authors, publication year, publication type, and country of the first author), research aims, developmental concepts, methods (e.g., research methodology and design, research setting, procedure and conditions, material, outcome measures, and type of CRI), child population characteristics (e.g., sample size, age, and socioeconomic background), robot characteristics (e.g., robotic platform, developer, exposition, physical CRI contact, purpose of use, form, appearance, autonomy, and behavior), reported gaps and limitations, research findings and conclusions. We exported the charted data from Covidence to Excel. All charted data is available at OSF (doi.org/10.17605/OSF.IO/WF48R).

### Stage 5: Collation, summarizing, and reporting results

The reviewed studies are summarized, reported, and discussed in line with the fifth stage of Arksey and O'Malley's scoping review framework in the following sections. We classified the studies based on the type of developmental concepts they involved.

## Results

### Search results

Overall, we identified 1747 studies from all database searches. After removing duplicates, and screening titles and abstracts, we screened 187 full texts for eligibility. Out of these, 158 studies were excluded. Finally, we included 29 studies in the review. Fig 1 shows the details of the search results and the study selection process in the PRISMA flowchart diagram [55].

### General characteristics

S1 Table provides an overview of all reviewed studies, including general characteristics, research methods, aims, sample characteristics, the robotic platform and other measures used, and a summary of the main findings and conclusions. There were 25 journal articles, three conference papers, and one magazine article. None of the studies were preprints. Studies were published between 1991 to 2023, and the research activity slightly grew over the past three decades (Fig 2).

The authors came from different countries, and most studies were conducted in Japan, followed by the United States and Canada (Table 2).

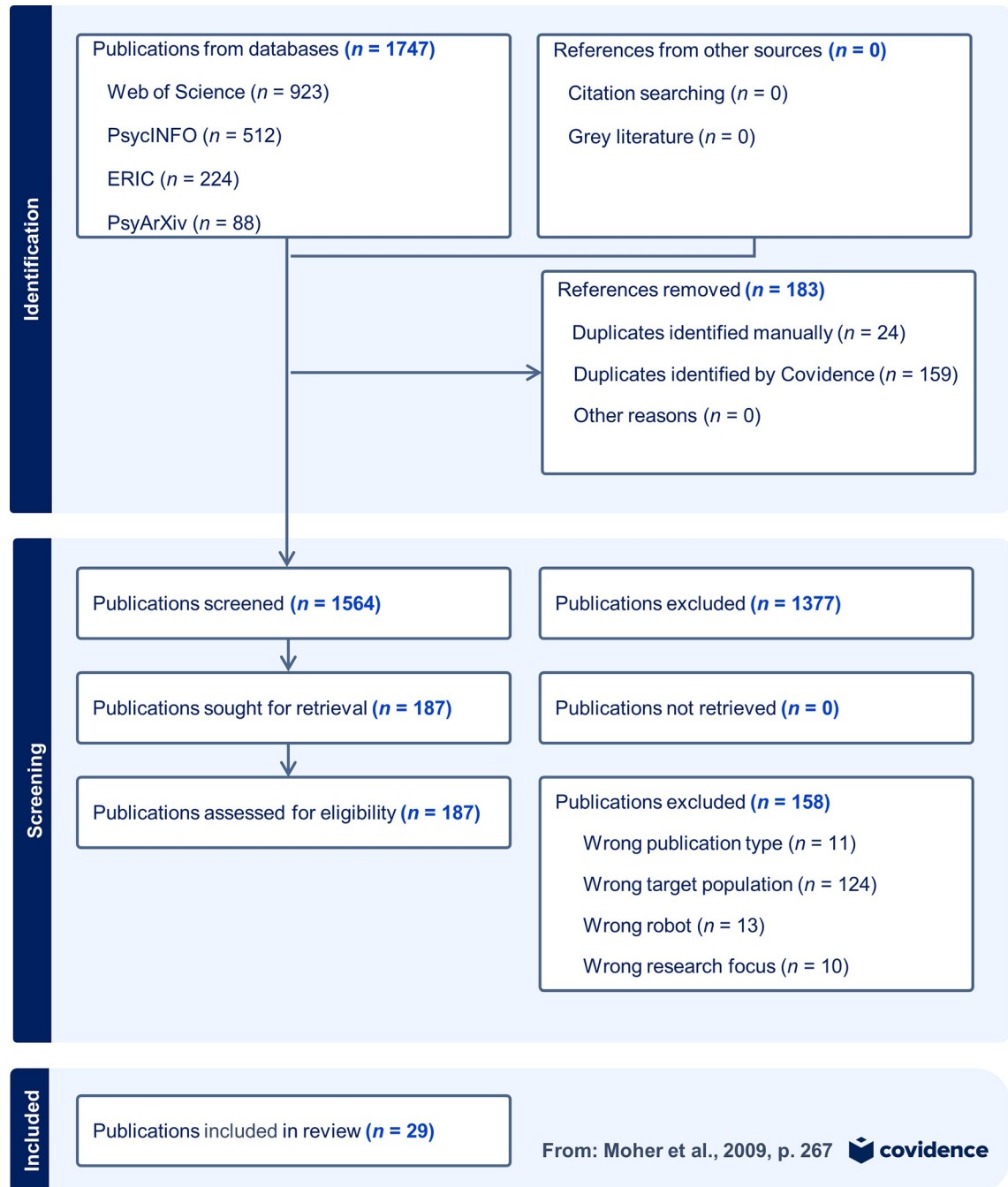

**Fig 1. PRISMA 2009 flowchart diagram.** The study selection process, including procedures of identification, and screening of studies. Studies were excluded based on a fixed order of exclusion reasons, including only the first incident of an unmet reason in this diagram.

## Research methods

Almost all studies ($n = 25$) used quantitative methodology, while only two studies used qualitative methodology and one used a mixed approach. Twenty-five of the studies used an experimental design, while the remaining four used a descriptive, correlational, case study, or ethnomethodology design. Twenty-four studies were conducted in a laboratory or in a

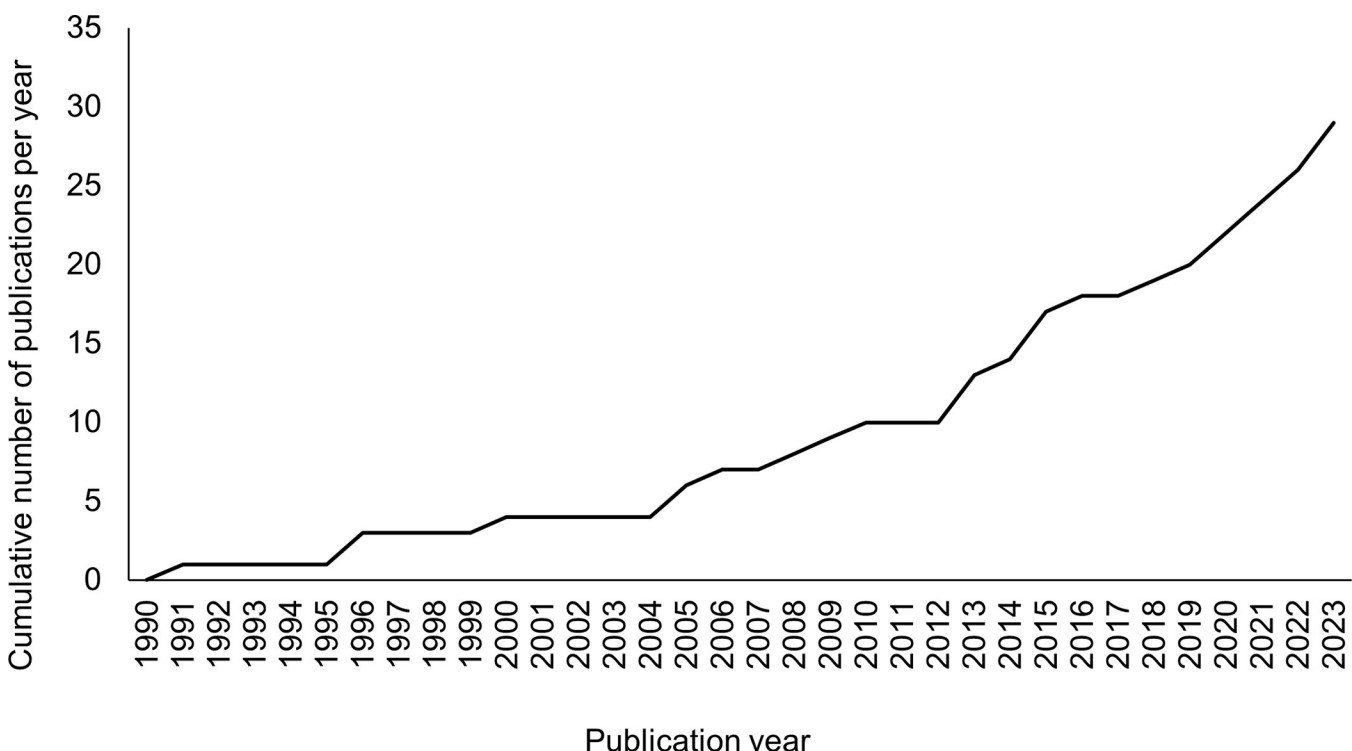

**Fig 2. Studies per year.** The cumulative number of studies per year between 1990 to 29. May 2023.

controlled laboratory setting. Two studies were conducted in ecological settings, such as classrooms. The remaining three studies were conducted in different locations, one study in a naturalistic setting at a science museum, and two studies used various locations (i.e., laboratory, ecological and/or naturalistic location).

## Child characteristics

The final sample sizes of the studies ranged from 6 to 230 participants, with the ages of participants ranging from 2 to 35 months. While some studies [56–62] included participants older than the target age, this review only focuses on findings related to children in the target age group. Twenty studies included toddlers who were 12 months or older, while seven studies included infants under 12 months. Five studies reported the socioeconomic status of the families [63–67], all belonging to the middle-class. For more details about the samples, see S1 Table.

**Table 2. Country distribution.** Countries of the lead authors (*N* = 29).

| Country | *n* |
|---|---|
| Japan | 8 |
| Taiwan | 1 |
| Italy | 3 |
| Romania | 1 |
| United Kingdom | 2 |
| Canada | 5 |
| United States | 6 |
| Australia | 3 |

## Robot characteristics and interaction types

We identified 16 social robots (Table 3 and Fig 3), most having a humanoid appearance (*n* = 24), whereas the remaining were animal-like (*n* = 4) and a ball-shaped robot (*n* = 1). The robots used were Robie Sr., Robovie, Robovie2, NAO, Dr. Robot Inc, HOAP-2, RUBI, RUBI-6, iRobiQ, Sphero, RepleeQ2, MyKeepon, Bee-Bot, 210 AIBO, MiRoE, and Opie. Robovie (versions 1 and 2) was most frequently used (*n* = 8). Most robots were pre-programmed to perform specific behaviors to examine children's responses to these acts (*n* = 24), such as making eye contact or gazing in the direction of an object [e.g., 68], or performing specific actions with objects [e.g., 62]. Two studies used autonomous robot dogs that acted by themselves and reacted to the children's behavior [60, 61]. Additionally, some [57, 58, 69] exposed children to robots that were autonomous or pre-programmed at different phases of the experiment.

In most studies, the robots were present in the same physical location as the child (*n* = 18), whereas the remaining robots were presented in video (*n* = 11). In most cases, the child-robot interaction did not involve any physical contact with the robot (*n* = 19). A total of 34 experiments were conducted in the 29 reviewed articles in which children were exposed to robots in some way. Most commonly, the robot was exposed to the child in a one-to-one interaction or situation (*n* = 20), including both live interactions and passive observations without social exchange. The remaining were bystander interactions (*n* = 5), where the child observed the robot interact with someone else, children-robot interactions in groups (*n* = 4), or a mixture of different interaction types (*n* = 5).

## Outcome measures and other instruments and material

Details of the outcome measures are presented in the S1 Table. The most frequent measure in the studies was children's looking behavior during stimuli presentation (*n* = 12). Looking behavior was measured using different instruments, such as eye tracking methods, video recordings captured by cameras, or observational notes. Various techniques were used to analyze looking behavior, such as visual habituation, preferential looking, violation of expectation, and anticipatory looking. Another common measure was children's imitation behavior assessed in imitation tests by analyzing the performance of target actions (*n* = 7).

## Research focus, key findings, and conclusions

The studies focused on several social and cognitive skills that we clustered into 4 main categories (Table 4). The key findings and conclusions of all studies are presented in the S1 Table.

**Animacy understanding.** Seven studies investigated children's understanding of animacy (Table 4). They examined how children classify robots as animate or inanimate based on their appearance [77, 91], movements [81], and interactive behaviors [60, 61, 82, 91], using both humanoid and animal-like robots (Table 3 and Fig 3). The findings were diverse, with children sometimes perceiving robots as more like living beings when the robots had a highly human-like appearance [77] or behaved contingently [82, 91, 92]. For example, infants aged 6 to 14 months did not differentiate between a highly human-like android and a human, viewing both as animate, but they recognized the difference between a human and a mechanical-looking robot (Fig 3) [77]. Contingency behavior influenced children's animacy understanding, with children's reactions to robots varying depending on the robots' contingency [82, 92]. Children aged 9 to 17 months who observed contingent interactions between a robot and a human were more likely to perceive the robot as a social being, suggesting the importance of responsive behavior in animacy perception [82, 92]. Nine- and twelve-month-old infants showed different expectations for human and robot movement, demonstrating increased negative affect when robots moved autonomously, suggesting that infants might consider robots inanimate

**Table 3. Robots used in the studies.** H = humanoid; NH = non-humanoid; *n* = number of studies using a given robot.

| Robot | Developer | Purpose | Form | Appearance | n | Representative studies |
|---|---|---|---|---|---|---|
| Robie Sr. | Radio Shack | Animacy concept, early conversational skills | H | Small toy robot with a head, ears, eyes, and a mouth. Wore a sweater/T-shirt and a cap/hat. Mounted on a wheeled base with a single unit body, arms, and hands. | 4 | [63–65, 81] |
| Robovie | ATR Media Information Science Laboratories; ATR Intelligence Robotics Laboratory | Action understanding (e.g., gaze following, goal attribution, attribution of intention to failed actions) | H | Large robot with a moveable head, eyes with pupils, body, torso, arms, and hands. Mounted on a wheeled base. | 7 | [68, 82–87] |
| Robovie2 | Hiroshi Ishiguro Laboratories | Animacy concept, early conversational skills, action understanding (e.g., gaze following) | H | Large robot with a movable head, movable eyes with pupils, body, torso, arms, hands, and fingers. Wore white gloves. | 2 | [72, 88] |
| NAO | SoftBank Robotics; Aldebaran | Motor imitation, contingency learning | H | Medium-sized robot with a head, mouth, LED-eyes, body, torso, shoulders, arms, hands, fingers, legs, and feet. | 3 | [69, 89, 90] |
| Dr. Robot Inc. | NR | Action understanding (e.g., gaze following) | H | Medium-sized robot mounted on a wheeled base, with body, torso, fixed arms, moveable head, mouth, and eyes with pupils. Wore a red shirt. | 1 | [66] |
| HOAP-2 | Fujitsu Laboratories | Action understanding (e.g., gaze following) | H | Medium-sized robot with body, torso, arms, legs, hands that open/close, pan-tilt moveable head, black-circled eyes (rims of cameras), and a nose. | 1 | [67] |
| RUBI | NR | Animacy concept | H | Large robot with body, torso, arms, hands, head, eyes (cameras), a fixed nose, and a fixed mouth. Equipped with a computer screen on its torso. | 1 | [91] |
| RUBI-6 | Movellan et al., (2009, 2005); Tanaka et al., (2006) | Action imitation with objects | H | Medium-sized robot with body, torso, moveable arms, pincer hands, moveable head with an Apple iPad mini displaying an animated cartoon face with eyes, pupils, nose, mouth, and eyebrows. Mounted on a base. Equipped with a computer screen on its torso. | 2 | [58, 62] |
| iRobiQ | Yujin Robots, Yujin Robot Co., Ltd | Reading skills and interest | H | Medium-sized robot mounted on a base, with body, torso, moveable arms, moveable head with eyes and mouth. The face has LEDs and sounds. | 1 | [56] |
| Sphero | NR | Physical play and emotions | NH | Small, white-colored robotic ball. Blue drawing of a head with eyes. | 1 | [57] |
| ReplieeQ2 | Osaka University and KOKORO Co. Ltd., Japan | Animacy concept | H | 1) Android: Human-like head with black hair, a face with silicone skin, eyes, black eyebrows, a nose, and a mouth with lips. 2) Robot: Wore a plastic mask with a human-like appearance. Has eyebrows, fixed black eyes, a nose, and a mouth with lips. Both robots have bodies with necks, shoulders, arms, hands, fingers, and legs. | 1 | [77] |
| MyKeepon | Kozima, Nakagawa, and Yano (2004) | Animacy concept | NH | Small yellow snowman-shaped and creature-like robot in soft silicone rubber. Head with fixed eyes with pupils. Mounted on a base. | 1 | [92] |
| Bee-Bot | TTS Group Ltd, 2021 | Computational thinking, programming, and coding skills | NH | Small and bee-like robot with black and yellow stripes on its body. Colorful buttons on top. Head with a fixed mouth and fixed eyes with pupils. | 1 | [59] |
| 210 AIBO | Sony | Animacy concept | NH | Small and dog-like robot with a head, eyes, nose, ears, legs, and a tail. Metallic form in black color. | 1 | [61] |
| MiRoE | Consequential Robots | Animacy concept | NH | Small and dog-like robot with a head, ears, eyes with pupils, moveable eyelids, body, neck, two wheeled legs. Wore a collar. | 1 | [60] |
| Opie | NR | Action imitation with objects | H | Large robot with an upper body, a head, animated eyes with pupils and eyelids, neck, torso, shoulders, arms, and hands. A head with a black-colored screen face. | 1 | [93] |

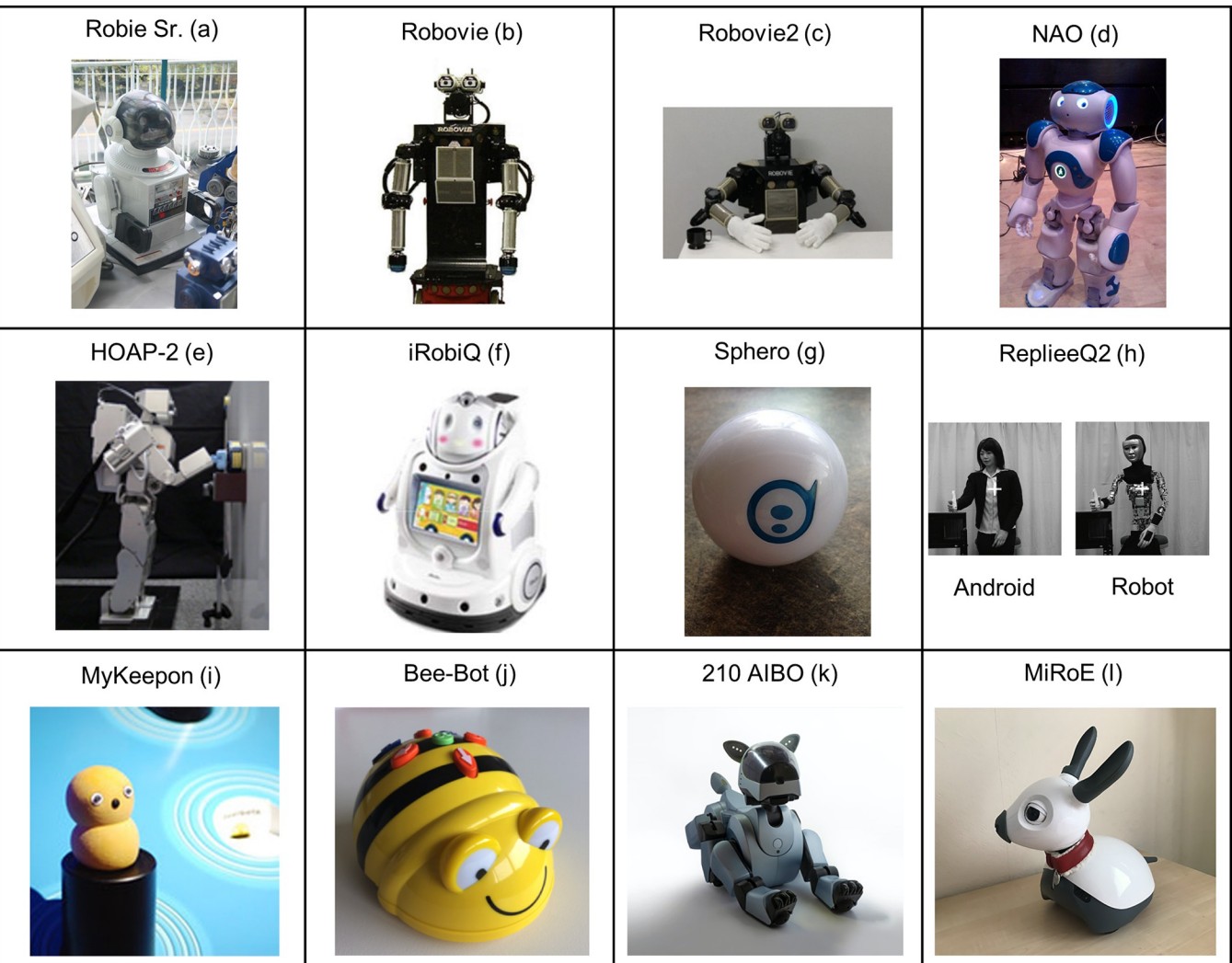

**Fig 3. Most of the robots in the review.** Images b, c, e, f, h, j, k, and l are modified cropped versions of the original work. Original images are licensed under CC-BY. For the robots Dr. Robot Inc., Opie, RUBI, and RUBI-6, we could not find images with a CC-BY (or similar) license. The Android and mechanical configurations of the same robot are shown in image (h). The image sources are: a) [70]; b) [71]; c) [72]; d) [73]; e) [74]; f) [75]; g) [76]; h) [77]; i) [78]; j) [79]; k & l); [80].

regardless of self-generated motion [81]. Studies with robot dogs showed that children differentiated between robotic dogs and toy dogs, but they did not necessarily view the robotic dog as a living animal [60, 61]. However, they did engage with the robotic dog in a manner

**Table 4. Research focus in the studies.** The other category includes the concepts of computational thinking ($n = 1$), reading interest and skills ($n = 1$), and physical play and emotions during robot interaction ($n = 1$).

| Social and cognitive concepts | Frequency | Representative studies |
|---|---|---|
| Animacy concept | 7 | [60, 77, 81, 82, 91, 92, 94] |
| Action understanding | 10 | [66–68, 72, 83–88] |
| Imitation | 6 | [58, 62, 69, 89, 90, 93] |
| Early conversational skills | 3 | [63–65] |
| Other | 3 | [56, 57, 59] |

suggesting that they perceived it as a social partner [60, 61]. Observations of 12- to 24-month-old toddlers' long-term interactions with a social robot indicated that they perceived the robot as a social partner [91]. The robot's interactivity, appearance, and inscriptions of gender and social roles influenced toddlers' attribution of animacy [91]. One study discussed anecdotal observations suggesting that toddlers may ascribe animacy to robots based on reciprocal vocalizations and social behaviors, such as inviting the robot to dance or apologizing to it after accidental contact [63]. Two studies connected children's concepts of animacy with their understanding of actions, particularly goal-directed and contingent actions [77, 91], which will be discussed in the section below on action understanding.

**Action understanding.** Ten studies used humanoid social robots to examine children's understanding of various actions (Tables 3 and 4), including referential actions [66, 67, 72, 84–86], goal-directed actions [83, 87, 88], and intentions behind failed actions [68]. Action understanding refers to the ability to recognize and respond appropriately to other's actions, infer the goals of actions, and detect the intention underlying the actions [95].

Studies on referential actions [66, 67, 72, 84–86] showed that children aged 10 to 18 months can follow the gaze of humanoid robots, but their understanding of the robot's intentions varied. For example, 12-month-olds respond to robot gaze, and it is not just an attentional reflex to its head movements [84], but they do not anticipate object appearance following robot gaze as they do for humans [84, 85]. Similarly, one study [72] found that 17-month-olds more frequently followed the human gaze than the robot gaze, suggesting that toddlers did not understand the referential intention of the robot's gaze. Yet, toddlers may still understand the robot's referential intentions, such as when the robots provide verbal cues during object learning [66, 86] or when the robot has previously engaged socially with adults [67]. Studies on goal-directed actions [83, 87, 88] showed that infants from 6.5 months could identify the goals of a humanoid robot as it is moving towards a goal destination, and they evaluate whether the robot is performing the most efficient path to reach its goal [83]. However, they do not attribute goals to a featureless box, suggesting that the human-like appearance of an agent influences infants' reasoning about an agent's actions [83]. Moreover, 13-month-old toddlers did not expect cooperative actions between humans and robots, even with social cues present [87]. By 17 months, toddlers showed signs of predicting the goal-directed reaching actions towards a target of both humans and humanoid robots, indicating an understanding of goal-directed behavior irrespective of the agent [106]. Finally, toddlers aged 24 to 35 months recognized the intention behind a robot's failed attempts to place beads inside a cup, but only when the robot made eye contact [68].

**Imitation.** Social robots were used to study two kinds of imitation in young children, i.e., their ability to learn by observing and imitating others [96]. Half of the studies focused on infants aged 2–8 months and their imitation of the humanoid robot's bodily movements, also known as motor imitation, and contingency learning in a face-to-face interaction [69, 89, 90]. Although 2- to 5-month-olds paid more attention to the robot when it moved, only 6- to 8-month-olds imitated its motor movements and demonstrated contingency learning [69, 89, 90]. The remaining studies investigated 1- to 3-year-old toddlers' imitation of a robot's actions with objects, such as assembling a rattle and shaking it to make a sound [58, 62, 93]. The studies found that toddlers imitate both physically present [58] and on-screen robots [62] and that their imitation of robots increased with age [58, 62]. Toddlers who interacted more with the robot prior to the imitation test were more likely to imitate it [58], though they still imitated humans more frequently [58, 62]. Moreover, toddlers' imitation from on-screen demonstrations of a human experimenter performing actions is not facilitated by presenting such videos embedded in robots behaving socially [93].

**Early conversational skills.** Three studies used a toy robot to investigate early conversational skills in toddlers (Tables 3 and 4). The robot provided constant verbal stimulation through an in-built speaker. By using a robot, the researchers aimed to eliminate potential confounding nonverbal cues (e.g., gaze, gestures) inevitably present in human conversation that could affect toddlers' responses [63–65]. For 24-month-olds, when the robot reciprocated toddlers' utterances by repeating and expanding the topic, it led to more topic-maintaining conversation and increased linguistically mediated social play [63]. Moreover, 24-month-olds recognized when the robot's responses were semantically relevant and on-topic, and in these situations, toddlers were more likely to continue and expand the conversational topic compared to when the robot was off-topic [64]. Older toddlers, aged 27 and 33 months, demonstrated an understanding of pragmatic quantity rules in conversations by responding appropriately to specific and general queries when conversing with the robot [65].

**Other concepts and related findings.** The remaining studies used various social robots (Table 3) to examine: reading ability [56], computational thinking programming, coding skills [59], and physical play and emotional responses [57]. For more details about these studies, see the S1 Table.

## Gaps and challenges

To address our third research question, we summarize gaps and challenges in using social robots as a research tool reported by the authors of the studies in the review. The most reported gaps by the authors were related to children's familiarity with robots, testing the effect of specific robot appearance and/or behavior cues, the design of the robot, and testing across different settings. Many studies [58, 62, 72, 82, 85, 87, 88] discussed that future work should investigate whether children's familiarity with robots might influence their understanding of and response to robots. For example, Okumura discusses [85] that infants might have stronger expectations for referential cues, such as gaze, from humans rather than robots due to their familiarity with human interaction. Moreover, future studies should investigate whether children's increased exposure to robots can enhance their ability to understand and respond to a robot's referential communication [85]. Several studies suggest that further research should investigate how a robot's physical appearance and behavior impact children's perception, comprehension, and learning from robots [66, 81–83, 85, 87]. For instance, Okumura et al. [86] suggest that future research should examine whether verbal cues provided by robots influence infants' object learning. Regarding gaps related to robotic design, one study [92] elucidated that robotic developers should aim to make robots that can interact autonomously without interference from a human operator. Related to the robot's design, Peca and colleagues [92] propose that future work should try to make robots that can interact autonomously with the child without the need for an operator. Most of the studies were conducted in experimental settings, and some studies [69, 72] suggest that future work should examine child-robot interactions in more naturalistic settings.

Most studies (*n* = 24) reported some challenges or limitations related to using social robots as a research tool. Many studies (*n* = 10) reported challenges related to the robot's design, such as issues related to its appearance and functionality. For example, additional human operators are required in the experimental procedures due to the technical constraints of the robots, difficulty in making the robots' movements resemble human movements, or challenges with using robots in live tasks because robots fail to provide the stimuli correctly or do not respond appropriately during interactions. Several studies (*n* = 7) reported children having challenges understanding the robot, such as its actions, communicative cues, and underlying intentions. Relatedly, some studies discussed that children's lack of familiarity and experience with robots

**Table 5. Reported challenges in using robots as a research tool in the included studies.** The category "no limitations reported" refers to studies that have not reported any challenges relevant to using social robots as a research tool.

| Challenges reported | Frequency | Representative studies |
|---|---|---|
| Child | | |
| Fear of robot | 3 | [58, 81, 93] |
| Novelty of robot | 4 | [66, 72, 84, 87] |
| Understanding the robot | 7 | [58, 66, 69, 72, 85–87] |
| On-task engagement | 5 | [58, 59, 87, 89, 90] |
| Sample bias | 1 | [65] |
| Robot | | |
| Design | 10 | [58, 61, 62, 66, 67, 72, 77, 91, 92] |
| Cost | 2 | [56, 69] |
| Safety hazards | 1 | [62] |
| Stimuli presentation | 2 | [66, 77] |
| Research design | | |
| Ecological validity | 3 | [63, 69, 88] |
| Chosen design | 1 | [92] |
| Operationalizing | 2 | [61, 92] |
| Setting or set-up | 3 | [60, 85, 90] |
| No limitations reported | 5 | [57, 64, 68, 82, 83] |

may contribute to difficulty understanding them and make them more distracting ($n = 4$). Several studies ($n = 5$) reported children experiencing challenges with task focus, including little or too much interest in the robot, irritability during robot inactivity, or children being distracted and leaving the task activity. Some studies ($n = 3$) discussed ecological validity issues, such as the generalization of findings across settings and with specific robots to other robot types or humans. Relatedly, we noticed that few studies used control groups with human or non-human agents for the robots they used, and there is limited discussion on the absence of these controls. An overview of commonly reported challenges is presented in Table 5.

## Discussion

This scoping review is a novel contribution to the field as it is the first to systematically cover the breadth of the literature on how social robots have been used in early development research to investigate social and cognitive development. Our review provides an overview of general characteristics, methods, research focus, findings, and the reported gaps and challenges when social robots are used in early developmental research. Previous systematic reviews and scoping reviews have focused on using social robots with older children in other settings, such as in education [97], supporting autism development [98–102], or various health care contexts [103–106]. Although we maintained the wide approach of a scoping review, we found that an overarching research focus in the reviewed literature was to determine if social robots can act as social partners for young children. According to this literature, children sometimes classify social robots as social partners and can interpret the social cues and actions of robots in certain situations. Thus, the studies demonstrate the potential of using various social robots in early developmental research, but do not suggest that social robots can replace humans in research settings.

### General characteristics and methods

We found that the use of social robots in early development research is a small research field, and we found 29 studies for the review. Most studies were quantitative with experimental

designs and conducted in controlled laboratory settings, in which the children were exposed to the robots in a one-to-one situation. Few studies used qualitative methodology [59, 60, 91], and only one study [91] observed child-robot interactions in a long-term context. Most robots were humanoid and pre-programmed to perform a specific social behavior of interest. We had a broad definition of social robots, including robots that fit typical descriptions of social robots, such as Robie Sr., Robovie, Robovie2, NAO, Dr. Robot Inc., HOAP-2, RUBI, RUBI-6, iRobiQ, ReplieeQ2, MyKeepon, 210 AIBO, MiRoE, and Opie (Table 3 and Fig 3). However, we also found robots not typically considered social robots, such as the robotic ball Sphero and Bee-Bot (Table 3 and Fig 3). Notably, the robots used in the studies varied in their level of advancement. Some were relatively simple and immobile, like the Robie Sr. robot, while others were capable of autonomous action, such as the NAO robot (Table 3 and Fig 3). Naturally, some of the more advanced robots were unavailable when the first studies were conducted, and therefore, we found that more simplistic robots were used in the studies that were first published.

## Research focus and key findings

Our review shows research trends in using social robots to study social and cognitive concepts such as animacy understanding, action understanding, imitation, and early conversational skills. Some studies also used robots to examine reading abilities, computational thinking, and emotions. We found that most studies focused on whether children classify robots as social partners to interact with and acquire information from or whether humans are a privileged source of information at these developmental stages [58, 60, 62, 66–69, 72, 77, 81–94]. Only a few studies [63–65] used robots to provide more constant stimuli instead of humans, with a main focus on the developmental concepts examined. Furthermore, some had an additional focus on the application of robots [56, 59, 60], such as the therapeutic potential of robot dogs [60] or as a learning tool to improve reading [56]. Lastly, one study used a robot providing socially contingent behaviors to facilitate children's imitation learning from a human experimenter [93].

The limited number of studies means that caution is necessary when interpreting the findings. Furthermore, research findings from one age group cannot be generalized to others. However, some key findings indicate that infants are attentive to robots and can learn from them at an early stage of development in several situations. Thus, humans are not necessarily the only information source for young children. For instance, 2-month-olds tend to be more attentive to robots that move [90], while 6-month-olds imitate robots [69]. Furthermore, 6.5-month-olds can attribute goals to a robot's moving actions toward a specific destination [83]. Another key finding was that as children grow older, they show signs of becoming better at recognizing and interpreting the social cues provided by robots, and their learning from robots is enhanced. For example, 24- to 35-month-old showed early signs of attributing intentions to robots by detecting what a robot intended to do when it failed to put beads inside a cup [68]. Additionally, 1-to-3-year-olds were able to imitate a robot's actions with objects both on-screen and in real life, and imitation increased with age [58, 62]. Yet, in several situations, children in the reviewed studies did not understand the robots' social behaviors and were not able to learn from them [66, 72, 84, 85, 87, 90]. Taken together, toddlers and infants may view robots as social partners, attributing mental states to them like older children do [107–110]. Moreover, this literature provides information on the ages at which young children can socially engage with social robots.

Yet another key finding was that it was not just the appearance of social robots but also how the robots behave that plays an important role in how young children perceive, understand, and respond to them [56, 58, 63, 64, 67, 82, 86, 91]. Especially, contingency and interactivity

behaviors facilitated how the robots were understood. For example, when young infants observed another person talking to or contingently interacting with a robot, they tended to classify the robot as animate [82, 92], and they showed increased sensitivity to its social cues such as eye gaze [67]. Additionally, toddlers who interacted more with the robot prior to the imitation test were more likely to imitate it [58]. In conversations with robots, toddlers tended to stay more engaged in the conversation when the robot reciprocated their verbalizations and stayed on-topic [63, 64]. Moreover, adding more social factors to the robot, such as verbal cuinging, increases 12-month-old infants' ability to follow a robot's gaze to an object [86]. Relatedly, Csibra [111] proposes that it is not how an agent looks that is important for children to identify it as an agent, but how it behaves. It is possible that social robots having appearances and social behaviors like living beings blur the lines between living and non-living beings and that social robots are represented as a new ontological category in children. As a result, young children might perceive and treat these robots as social partners and not just machines. Relatedly, Manzi [88] et al. discuss robots with human-like characteristics might activate social mechanisms in young infants. Yet, in some cases, appearance and contingency behaviors were not enough to elicit an understanding of the robot's intention [66].

## Gaps and challenges

The authors reported several gaps and challenges related to using social robots in early developmental research. Most commonly, the authors reported that future work should investigate whether children's familiarity with robots impacts their responses. Although social robots possess human-like qualities and behaviors already familiar to the child, their novelty may result in different responses from children when compared to interactions with human agents. Frequently reported challenges were related to robot design. For instance, in some studies, a human experimenter had to accompany the robot during an experiment because of the technical constraints of the robots [66, 92]. Relatedly, Peca and colleagues [92] discuss that future work should aim to make robots that do not require human operators.

## Limitations

This scoping review is not without limitations. Although we conducted extensive searches across multiple databases, it is possible that some relevant studies were not included. Our inclusion criteria were limited to studies published in English, and we did not manually search reference lists to identify additional studies, which may have resulted in the exclusion of relevant studies. Furthermore, as scoping reviews do not typically aim to assess the quality of evidence, we did not perform a formal quality assessment of the studies included.

## Future directions

This review has allowed us to identify important directions for future research, primarily within developmental psychology but also in social robotics. Firstly, it is unclear how efficient social robots are when acting as agents in early developmental research. This is indicated by diverse findings related to how children classify them as animate or inanimate and how children interpret their social cues and behaviors. Notably, few studies used any human or non-human controls for robots. Thus, future studies should use other agent types in addition to robots to determine the efficiency of using social robots, humans, and other types of agents in early developmental research. Findings on what robot behaviors are crucial for young children may have implications for future work within social robotics when aiming to develop age-appropriate robots. Secondly, we found that multiple robots were rarely used within the same study, and thus, it is unclear if their findings generalize to other types of robots or if the

findings are specific to a particular robot type. Future work could use several robots to test generalizability across different robot types. Thirdly, most studies investigated child-robot interactions in highly controlled settings that do not easily generalize to other environments. Future work should investigate naturalistic interactions between children and robots, in which the robots respond to the child's behavior at the moment rather than being pre-programmed to do a specific task. Fourth, we noticed that the included studies rarely reported the reasons behind their choice of a specific robot type and the amount of time spent preparing the robot, such as learning to program it or having a skilled programmer do it. We suggest reporting such information to ease replication and to improve planning for future studies.

## Conclusion

Our scoping review of 29 studies shows a small and emerging field of using social robots to study social and cognitive development in infants and toddlers. We identified four main areas of focus: animacy understanding, action understanding, imitation, and early conversational skills. An important question in the field is whether young children perceive social robots as social partners or agents. Findings vary on how children classify and understand the behaviors of social robots. According to the studies, young children can, from an early age, pay attention to social robots, learn from them, and recognize their social signals, but not always. The studies suggest that certain robot behaviors, particularly those that are interactive and contingent, are critical for enhancing children's perception of robots as social entities. Moreover, it seems like children's understanding of robots improves with age. Our review indicates that even in infancy, social robots can be regarded as social partners, a perception that is essential in research settings that depend on social interaction. Consequently, our review highlights the need for careful selection of social robots that exhibit interactive and contingent behaviors to be effective in early developmental research. Furthermore, this review contributes knowledge on how children socially interact with and learn from non-human agents with rich social features. These insights are important for future studies within developmental psychology involving social robots and young children and future work within social robotics on designing appropriate robot behaviors to facilitate social interaction with robots in early childhood.

## Supporting information

**S1 Checklist. Preferred Reporting Items for Systematic reviews and Meta-Analyses extension for Scoping Reviews (PRISMA-ScR) checklist.**
(DOCX)

**S1 File. Search strategy.** Search queries and search terms used in the databases and preprint repository.
(DOCX)

**S1 Table. Overview of the included studies.**
(DOCX)

## Acknowledgments

We thank Torstein Låg, Senior Academic Librarian at the UiT The Arctic University of Norway, for support in developing search strategies.

## Author Contributions

**Conceptualization:** Solveig Flatebø, Vi Ngoc-Nha Tran, Catharina Elisabeth Arfwedson Wang, Lars Ailo Bongo.

**Data curation:** Solveig Flatebø.

**Formal analysis:** Solveig Flatebø.

**Investigation:** Solveig Flatebø, Vi Ngoc-Nha Tran, Lars Ailo Bongo.

**Methodology:** Solveig Flatebø.

**Project administration:** Solveig Flatebø.

**Supervision:** Vi Ngoc-Nha Tran, Catharina Elisabeth Arfwedson Wang, Lars Ailo Bongo.

**Visualization:** Solveig Flatebø.

**Writing – original draft:** Solveig Flatebø.

**Writing – review & editing:** Solveig Flatebø, Vi Ngoc-Nha Tran, Catharina Elisabeth Arfwedson Wang, Lars Ailo Bongo.

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
