## [Decision Letter · Decision Letter 0]

22 Mar 2024

PONE-D-24-04880Social robots in research on social and cognitive development in infants and toddlers: A Scoping reviewPLOS ONE

Dear Dr. Flatebø,

Thank you for submitting your manuscript to PLOS ONE. After careful consideration, we feel that it has merit but does not fully meet PLOS ONE’s publication criteria as it currently stands. Therefore, we invite you to submit a revised version of the manuscript that addresses the points raised during the review process.

We look forward to receiving your revised manuscript.

Kind regards,

Simone Varrasi

Academic Editor

PLOS ONE

Journal Requirements:

2. We note that Figure 3 includes an image of a participant in the study. 

3. Please remove your figures from within your manuscript file, leaving only the individual TIFF/EPS image files, uploaded separately. These will be automatically included in the reviewers’ PDF.

4. We note that this manuscript is a systematic review or meta-analysis; our author guidelines therefore require that you use PRISMA guidance to help improve reporting quality of this type of study. Please upload copies of the completed PRISMA checklist as Supporting Information with a file name “PRISMA checklist”.

Additional Editor Comments:

Both Reviewers recognized the high value of your manuscript. They reported some suggestions that do not question the methodological structure of your work, but yet they are important for the improvement of its quality and for the adherence to editorial guidelines. Please consider them as appropriate.

Reviewers' comments:

Reviewer's Responses to Questions

**Comments to the Author**

1. Is the manuscript technically sound, and do the data support the conclusions?

Reviewer #1: Yes

Reviewer #2: Yes

2. Has the statistical analysis been performed appropriately and rigorously? 

Reviewer #1: N/A

Reviewer #2: Yes

3. Have the authors made all data underlying the findings in their manuscript fully available?

Reviewer #1: Yes

Reviewer #2: Yes

4. Is the manuscript presented in an intelligible fashion and written in standard English?

Reviewer #1: Yes

Reviewer #2: Yes

5. Review Comments to the Author

Reviewer #1: Dear authors,

your scoping review about social robots in research on social and cognitive development in infants and toddlers is really interesting and scientifically useful.

It is fluid and well written.

Minor concerns are suggested:

From line 60, you deal with the scientific literature about the role of social robot as methodology of developmental psychology research. You could introduce this topic dealing with how social robots began to be used in assessment. These pioneering studies could help you: https://doi.org/10.1007/978-3-319-89327-3_8 ; https://doi.org/10.1007/978-3-319-96728-8_34 (in this case, the paper is focused on children with autism and intellectual disability, but it is really interesting);

Reviewer #2: Dear authors,

thank you for submitting your work to the journal PLOS ONE.

I reckon the study is very interesting.

I just found few details that may need to be checked.

1. Bibliography isn’t justified and the insertion of DOI would be appreciated.

2. Tables should have their description underneath, not above (e.g. tab. 3, 5, 6…)

3. Use grid lines, especially separating long lists. (S1 Table 1).

4. In which field would you like to expand the future perspective of the technique?

6. PLOS authors have the option to publish the peer review history of their article (what does this mean?). If published, this will include your full peer review and any attached files.

Reviewer #1: No

Reviewer #2: No

---

## [Author Response · Author response to Decision Letter 0]

22 Apr 2024

Dr Simone Varrasi 

Academic Editor

PLOS ONE

Dear Dr Simone Varrasi,

Thank you for your decision letter regarding our submission PONE-D-24-04880 “Social robots in research on social and cognitive development in infants and toddlers: A Scoping review”, and for inviting us to resubmit our manuscript after minor revisions. For clarity, throughout this letter, we use italics to mark your and the Reviewer’s comments and blue to mark our new text. 

We have carefully read both the reviews and your letter. We made a great effort to incorporate all the points offered in the letter and in the two reviews to improve our manuscript. We have also proofread the manuscript again and corrected minor typos and errors. Below is a detailed list of how we addressed all the Reviewers’ and your points. We hope that with this minor revision, our manuscript will be accepted for publication in PLOS ONE. Please see below how we addressed your and the reviewer’s feedback.

Sincerely,

Solveig Flatebø

PhD student

UiT The Arctic University of Norway

-

solveig.flatebo@uit.no

+47 95 48 46 03 

[Our response:]

We have double-checked, and our manuscript follows PLOS ONE’s style requirements, and the files are named correctly. 

2. We note that Figure 3 includes an image of a participant in the study. As per the PLOS ONE policy (http://journals.plos.org/plosone/s/submission-guidelines#loc-human-subjects-research) on papers that include identifying, or potentially identifying, information, the individual(s) or parent(s)/guardian(s) must be informed of the terms of the PLOS open-access (CC-BY) license and provide specific permission for publication of these details under the terms of this license. Please download the Consent Form for Publication in a PLOS Journal(http://journals.plos.org/plosone/s/file?id=8ce6/plos-consent-form-english.pdf). The signed consent form should not be submitted with the manuscript, but should be securely filed in the individual's case notes. Please amend the methods section and ethics statement of the manuscript to explicitly state that the patient/participant has provided consent for publication: “The individual in this manuscript has given written informed consent (as outlined in PLOS consent form) to publish these case details” If you are unable to obtain consent from the subject of the photograph, you will need to remove the figure and any other textual identifying information or case descriptions for this individual.

[Our response:]

All images in Figure 3 are licensed under CC-BY and can freely be used by others, including image “h” of the ReplieQ2 robot. Image “h” displays two configurations of the “female” ReplieQ2 robot developed by Kokoro and Osaka University and Advanced Media, Inc. The leftward picture of the robot is the original Android configuration, whereas the picture to the right is the same robot stripped down to its underlying mechanical look. Therefore, since this is a picture of a robot and not a human participant, we have not removed picture “h” in Figure 3. However, we acknowledge that the android can easily be mistaken for a human, and we have therefore added the following sentence in the figure’s note to clarify (lines 250-251 in the clean manuscript): 

[Our new text:]

[…]. The Android and mechanical configurations of the same robot are shown in image (h). 

3. Please remove your figures from within your manuscript file, leaving only the individual TIFF/EPS image files, uploaded separately. These will be automatically included in the reviewers’ PDF.

[Our response:]

Thanks for bringing this to our attention. All figures within the manuscript files have been removed, and the individual TIFF image files have been uploaded separately. 

4. We note that this manuscript is a systematic review or meta-analysis; our author guidelines therefore require that you use PRISMA guidance to help improve reporting quality of this type of study. Please upload copies of the completed PRISMA checklist as Supporting Information with a file name “PRISMA checklist”.

[Our response:]

We have now uploaded the PRISMA checklist as Supporting Information with the correct file name, “S1_File_PRISMA_Checklist,” as requested. The PRISMA checklist was also uploaded in the first submission, but it did not have the correct file name.

[Our response:]

We have reviewed our reference list, and it is complete and correct. We have not cited any retracted papers. 

Reviewer #1: Dear authors, your scoping review about social robots in research on social and cognitive development in infants and toddlers is really interesting and scientifically useful. It is fluid and well written.

Reviewer #2: Thank you for submitting your work to the journal PLOS ONE. I reckon the study is very interesting.

[Our response:]

We thank both Reviewers for this positive feedback.

Reviewer #1: From line 60, you deal with the scientific literature about the role of social robot as methodology of developmental psychology research. You could introduce this topic dealing with how social robots began to be used in assessment. These pioneering studies could help you: https://doi.org/10.1007/978-3-319-89327-3_8 ; https://doi.org/10.1007/978-3-319-96728-8_34 (in this case, the paper is focused on children with autism and intellectual disability, but it is really interesting)

[Our response:]

We thank the reviewer for introducing us to these studies on robots' implications in psychological assessments. As the reviewer suggested, we read the papers linked to and read about the topic. In the Introduction, we added sentences that point to the importance of the topics raised by the Reviewer in connection with our study (lines 61-63 in the clean manuscript).

 [Our new text:]

Some pioneering studies have also demonstrated that social robots can contribute to cognitive assessments of elderly people and children with autism [32, 33]. 

[Our response continued:]

Moreover, we added the reference https://doi.org/10.1007/978-3-319-89327-3_8 to our discussion about the advantages of using social robots in research in the Introduction (lines 75-76 in the clean manuscript). 

[Our new text:]

[…]. Firstly, they provide a level of control and consistency that can be challenging to achieve with human experimenters [32, 44].

Reviewer #2: 1. Bibliography isn’t justified and the insertion of DOI would be appreciated.

[Our response:]

Thank you for bringing this matter to our attention. We have now applied the PLoS reference style, which includes the DOI of all papers. 

Reviewer #2: 2. Tables should have their description underneath, not above (e.g. tab. 3, 5, 6…)

[Our response:]

We have now checked all tables in the manuscript and supporting information and changed them so that they meet the table requirements (https://journals.plos.org/plosone/s/tables). We have also corrected the tables so that the table labels and titles are presented in bold font underneath them. 

Reviewer #2: 3. Use grid lines, especially separating long lists. (S1 Table 1).

[Our response:]

We have applied grid lines to all our tables, also in the long lists in the S1 Table 1 and in Table 4. 

Reviewer #2: 4. In which field would you like to expand the future perspective of the technique?

[Our response:]

Thanks for raising this important issue. We believe that the manuscript primarily falls under the field of developmental psychology, but it also has implications for the field of social robotics. The manuscript highlights crucial features of robots that are significant for young children and must be considered when developing age-appropriate robots in social robotics. However, the main topic in the reviewed literature is whether infants and toddlers perceive social robots as social partners, which are fundamental research questions belonging to the field of developmental psychology. We have highlighted this issue within the manuscript by adding several new sentences in the Future directions (lines 503-504 and 510-512) and by rewriting the last sentences in the Conclusion (lines 539-542 in the clean manuscript) to clarify which fields we would like to expand the future perspective. 

[Our new text:]

This review has allowed us to identify important directions for future research, primarily within developmental psychology but also in social robotics. (Lines 503-504 in the clean manuscript).

[…] Findings on what robot behaviors are crucial for young children may have implications for future work within social robotics when aiming to develop age-appropriate robots. (Lines 510-512 in the clean manuscript).

These insights have implications for future studies within developmental psychology involving social robots and young children and future work within social robotics on designing appropriate robot behaviors to facilitate social interaction with robots in early childhood. (Lines 539-542 in the clean manuscript).

---

## [Editor Report · Decision Letter 1]

30 Apr 2024

Social robots in research on social and cognitive development in infants and toddlers: A Scoping review

PONE-D-24-04880R1

Dear Dr. Flatebø,

We’re pleased to inform you that your manuscript has been judged scientifically suitable for publication and will be formally accepted for publication once it meets all outstanding technical requirements.

Kind regards,

Simone Varrasi

Academic Editor

PLOS ONE
---

## [Editor Report · Acceptance letter]

2 May 2024

PONE-D-24-04880R1 

PLOS ONE

Dear Dr. Flatebø, 

I'm pleased to inform you that your manuscript has been deemed suitable for publication in PLOS ONE. Congratulations! Your manuscript is now being handed over to our production team.

Kind regards, 

on behalf of

Dr. Simone Varrasi 

Academic Editor

PLOS ONE